# MetGEMs Toolbox: Metagenome-scale models as integrative toolbox for uncovering metabolic functions and routes of human gut microbiome

Preecha Patumcharoenpol[1], Massalin Nakphaichit[2], Gianni Panagiotou[3,4,5], Anchalee Senavonge[6], Narissara Suratannon[6], Wanwipa Vongsangnak[7,8]*

1 Interdisciplinary Graduate Program in Bioscience, Faculty of Science, Kasetsart University, Bangkok, Thailand, 2 Department of Biotechnology, Faculty of Agro-Industry, Kasetsart University, Bangkok, Thailand, 3 Systems Biology & Bioinformatics Group, School of Biological Sciences, The University of Hong Kong, Hong Kong S.A.R., China, 4 Department of Medicine and State Key Laboratory of Pharmaceutical Biotechnology, The University of Hong Kong, Hong Kong S.A.R., China, 5 Systems Biology & Bioinformatics Unit, Leibniz Institute for Natural Product Research and Infection Biology–Hans Knöll Institute, Jena, Germany, 6 Pediatric Allergy & Clinical Immunology Research Unit, Division of Allergy and Immunology, Department of Pediatrics, Faculty of Medicine, Chulalongkorn University, King Chulalongkorn Memorial Hospital, the Thai Red Cross Society, Bangkok, Thailand, 7 Department of Zoology, Faculty of Science, Kasetsart University, Bangkok, Thailand, 8 Omics Center for Agriculture, Bioresources, Food, and Health, Kasetsart University (OmiKU), Bangkok, Thailand

* wanwipa.v@ku.ac.th

## Abstract

Investigating metabolic functional capability of a human gut microbiome enables the quantification of microbiome changes, which can cause a phenotypic change of host physiology and disease. One possible way to estimate the functional capability of a microbial community is through inferring metagenomic content from 16S rRNA gene sequences. Genome-scale models (GEMs) can be used as scaffold for functional estimation analysis at a systematic level, however up to date, there is no integrative toolbox based on GEMs for uncovering metabolic functions. Here, we developed the MetGEMs (metagenome-scale models) toolbox, an open-source application for inferring metabolic functions from 16S rRNA gene sequences to facilitate the study of the human gut microbiome by the wider scientific community. The developed toolbox was validated using shotgun metagenomic data and shown to be superior in predicting functional composition in human clinical samples compared to existing state-of-the-art tools. Therefore, the MetGEMs toolbox was subsequently applied for annotating putative enzyme functions and metabolic routes related in human disease using atopic dermatitis as a case study.

## Author summary

Examining the metabolic function capability of microbiome is an important step in understanding the microbe-microbe and microbe-host communication. In this study, we

**Data Availability Statement:** All relevant data are within the paper, its Supporting Information files and from https://github.com/yumyai/MetGEMs.

**Funding:** WV received funding supports from Kasetsart University Research and Development Institute (KURDI) at Kasetsart University, Department of Zoology, Faculty of Science, as well as Omics Center for Agriculture, Bioresources, Food, and Health, Kasetsart University (OmiKU). PP received funding supports from Interdisciplinary Graduate Program in Bioscience, Faculty of Science, Kasetsart University and International Affairs Division (IAD), Kasetsart University. NS received funding supports from the National Science and Technology Development Agency (NSTDA) (Grant No. P-17-50648) and Ratchadapisek Research Funds (Grant No. CU-GR (S)_61_38_30_03) Chulalongkorn University. GP would like to thank the Deutsche Forschungsgemeinschaft (DFG) CRC/Transregio 124 "Pathogenic fungi and their human host: Networks of interaction", subprojects B5 and INF. The funders had no role in study design, data collection and analysis, decision to publish, or preparation of the manuscript.

**Competing interests:** The authors have declared that no competing interests exist.

developed MetGEMs (metagenome-scale models) toolbox, a strategic approach to infer metabolic functions from 16S rRNA gene sequences with overall aiming at annotating metabolic functions in the human gut microbiome. During the development, MetGEMs toolbox was tested and validated with shotgun metagenomic data. At the end, we used our toolbox to identify putative enzyme functions and metabolic routes associated to atopic dermatitis from the Thailand Pediatric Allergy Research Cohort. This work highlights the importance of metagenome-scale models as crucial framework for uncovering metabolic functions and routes of human gut microbiome.

This is a *PLOS Computational Biology* Methods paper.

## Introduction

Microbial community and diversity in human gut have demonstrated their effects on both health and disease [1–3]. Up to date, the gut microbiome studies through metagenomic analysis focus on finding a relationship between specific groups of bacteria and host's physiology [2, 4–6], as well as their functional interactions. Shotgun metagenomic sequencing is a key driving force for providing information about both taxonomic and functional gut diversities. It is used for the direct quantification of the functional profile in microbiome samples by directed sequencing all microbial DNA from samples [7, 8], nonetheless this approach is still limited to small-scale studies due to high cost and computational complexity. Alternatively, even though 16S rRNA gene sequencing approach does not sequence a specific functional gene directly, an inference of availability of specific genes and their functional role can be drawn. PICRUSt [9] was the first functional predictor to estimate potential functions in the microbial communities using sequenced 16S rRNA gene data integrated with draft microbial genomes from public databases. In addition to PICRUSt, more functional predictors were recently developed under the similar concept, such as PICRUSt2 [10], Tax4fun2 [11], Piphillin [12], CowPI [13], PanFP [14], PAPRICA [15], and BUGBASE [16]. These functional predictors took leverage on 16S rRNA gene sequences integrated with the draft microbial genomes where the automatic genome annotation process was usually used. It has been shown that the quality of reference genome and annotation has an impact on the predictive performance [11, 13], due to errors in the draft genomes, such as missing coding regions and ambiguous or false annotation [17–19]. With the advancement in systems biology, genome-scale models (GEMs) of human gut bacteria were reconstructed and stored in AGORA collections [20]. Recently, AGORA collections have been employed to investigate the metabolic reactions of gut microbiome across different studies [20–24]. However, there is no toolbox which has widespread and reproducible application for inferring metabolic functions from 16S rRNA gene sequences using GEMs as scaffolds.

Therefore, the aim of this study is to develop MetGEMs (metagenome-scale models) toolbox, an open-source application for inferring metabolic functions from 16S rRNA gene sequences with ultimate goal at annotating metabolic functions of human gut microbiome. MetGEMs toolbox was firstly developed by constructing MetGEMs networks, a generalized genome-scale model, using the AGORA collections [20] and the Human Microbiome Project (HMP) [25]. We then built and tested four different MetGEMs models to find the best computational method at metabolic function prediction. The MetGEMs toolbox's prediction capability was validated with the corresponding data derived from shotgun metagenomic sequencing

and PICRUSt2. Here, PICRUSt2 was selected since it has capabilities to predict metabolic functional potential of a community based on marker gene sequencing profiles, particularly it shows well-prediction performance on HMP data [10]. Subsequently, the MetGEMs toolbox was used for annotating putative enzyme functions related to allergic diseases i.e., atopic dermatitis from our Thailand Pediatric Allergy Research Cohort. The developed Met-GEMs toolbox serves as an integrative work for the metabolic functional analysis of microbial communities of the human gut microbiome using 16S rRNA gene sequences.

## Results and discussion

The MetGEMs toolbox was developed with the most up-to-date GEMs to infer metagenomic content from 16S rRNA gene sequences with overall focus on annotating metabolic function of human gut microbiome (Fig 1). This was done by leverage the gene content in GEMs and extrapolated with 16S rRNA gene sequences. MetGEMs toolbox, user's guide and sample data-set are publicly available for academic use at https://github.com/yumyai/MetGEMs.

### The assessed and evaluated GEMs

AGORA collections [20] containing 818 GEMs from human gut microbiome were initially collected. These 818 GEMs totally covered 1,470 KEGG Orthology (KO) identifiers (IDs) and 983 EC numbers across 226 genera and 690 species (S1 Spreadsheet). GEMs in the context of KO IDs and EC numbers across the taxonomic group (class) level were highly correlated with Pearson's correlation coefficient (PCC) of 0.99 (see Figs 2A and S1). Notably, the taxonomic-related bacteria at some classes tend to have similar functions, i.e. KO IDs and EC numbers, such as Coriobacteriia (549±65 KO IDs and 392±48 EC numbers) and Epsilonproteobacteria (583±40 KO IDs and 383±40 EC numbers) (Fig 2B). However, a different metabolic diversity exhibited at other classes, such as Bacilli and Actinobacteria, which have KO IDs of 707±161 and 693±132, as well as EC numbers of 458±107 and 458±95, respectively (Fig 2B). The high diversities of KO IDs and EC numbers identified in the example classes of Bacilli and Actino-bacteria in GEMs suggest that functional annotation at the high level of taxonomic groups (e.g., class, or even higher levels) might produce spurious predictions on the metabolic capac-ity. With this in mind, we developed MetGEMs network for genus-based metabolic functional annotation as shown below.

### The constructed MetGEMs network as reference database towards implementing MetGEMs toolbox

The MetGEMs network was designed as a reference database for inferring metabolic func-tional abundance of a certain taxonomic organism and/or group. Thus, a preliminary con-struction of MetGEMs network by KO IDs and EC numbers together with biochemical reactions at a certain genus was performed. To construct the MetGEMs network, four different computational approaches were applied, namely 1) Pan-Function, 2) Core-Function, 3) Pan-Weight-Function, and 4) Core-Weight-Function. Pan-Function considers all possible meta-bolic functions of every organism while Core-Function concerns only common metabolic functions of every organism. For Pan-Weight- and Core-Weight-Function, the taxonomic weight from HMP's gut microbiome was considered and combined with Pan- and Core-Func-tion (See Materials and Methods). Each approach used KO IDs and EC numbers existing in GEMs together with genome sequences related to GEMs for the MetGEMs network construc-tion. Among these four approaches, as expected the constructed MetGEMs networks are com-parable in context of coverage of KO IDs and EC numbers as listed in Table 1. Pan-Function has the highest coverage of 1,470 KO IDs and 983 ECs when compared to the others. This is

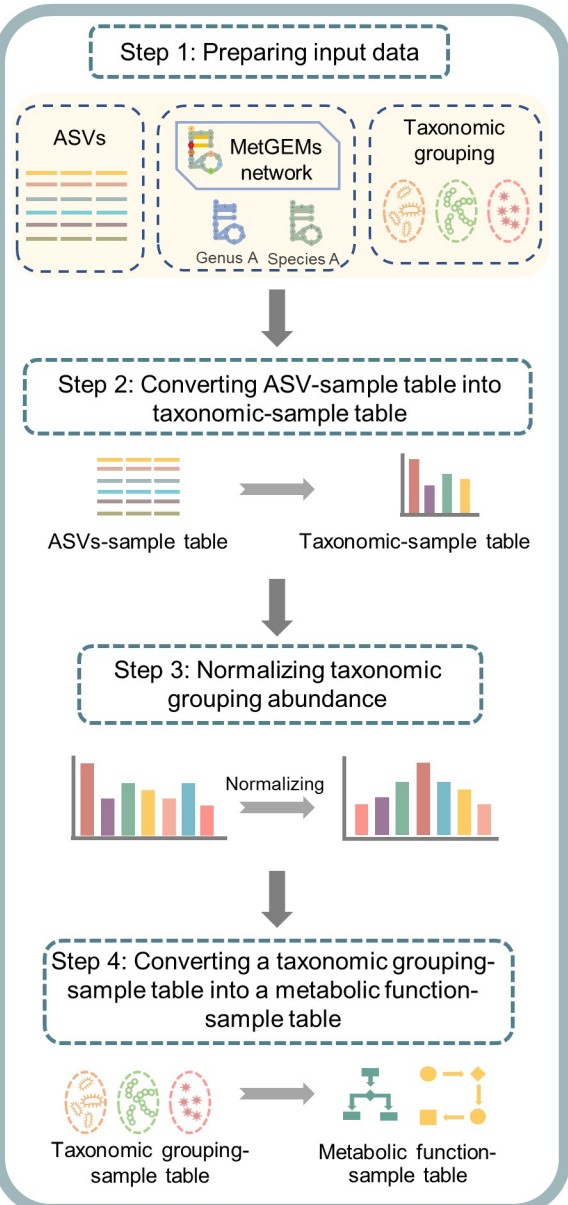

**Fig 1. A schematic workflow of a development MetGEMs toolbox.** The computational framework for the developed MetGEMs toolbox is divided into five different sections, namely 1) the assessed and evaluated GEMs, 2) the constructed MetGEMs network as reference database towards implementing MetGEMs toolbox, 3) the validated MetGEMs toolbox's prediction capability with shotgun metagenomic sequencing data, 4) using MetGEMs toolbox for assigning enzyme functions and relevant functional categories related human gut microbiome, and 5) annotating putative enzyme functions related in allergic disease using the MetGEMs toolbox as described in following.

possibly because Pan-Function considered all possible metabolic functions of every organism taking into the constructed MetGEMs network and therefore it contained the greatest number of metabolic functions. In contrast, Core-Function has the lowest coverage of 1,459 KO IDs and 968 EC numbers when compared to the others. This is because Core-Function captured only overlapped metabolic functions into the constructed MetGEMs network and these

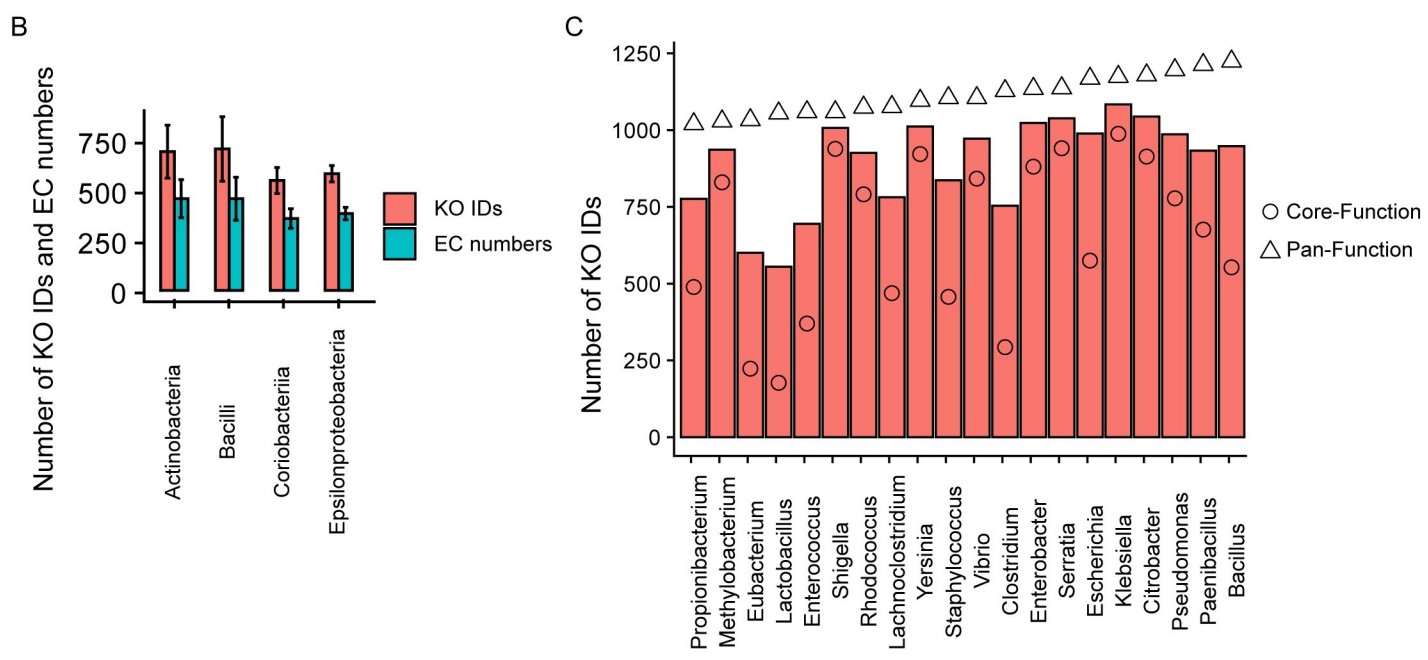

**Fig 2. A metabolic diversity of assessed most up-to-date GEMs.** (A) A Pearson's correlation analysis between KO IDs and EC numbers identified in the evaluated 818 GEMs across different taxonomic classes at a phylogenetic-tree scale. Full details of correlation analysis and goodness of fit between KO IDs and EC numbers by scatter plot can be seen in S1 Fig. Note: In this Figure, the blue and green lines indicate the numbers of KO IDs and EC numbers, respectively. PCC stands for Pearson's correlation coefficient. The phylogenetic-tree scale is built from 773 genomes with coding sequences available in AGORA collections visualized by iTOL [26]. (B) A bar graph explicates the selected taxonomic classes in context of the average numbers of KO IDs and EC numbers. (C) A bar graph illustrates a comparison of Core- and Pan-Function computational approaches in terms of the average numbers of KO IDs across different genera.

resulted in the smallest number of identified metabolic functions. In addition to Pan- and Core-Function, MetGEMs networks were also constructed by considering the taxonomic weight from HMP's gut microbiome. Indeed, Pan-Weight-Function and Core-Weight-Function are very similar to Pan- and Core-Function, respectively (Table 1). Further comparative analysis between Pan- and Core-Function are shown in Fig 2C. Interestingly, the Pan-Function provided very high number of KO IDs in a similar trend across different genera, while the Core-Function captured different numbers of KO IDs across different genera. In view of Core-Function, three genera e.g., *Lactobacillus* (177 KO IDs), *Eubacterium* (223 KO IDs) and *Clostridium* (293 KO IDs) clearly showed low numbers of KO IDs than the other genera in comparison to Pan-Function. On the other hand, *Klebsiella* had a high number of KO IDs (988 KO IDs) than the other genera. These evidences suggest that Core-Function could capture distinctive metabolic functions at an individual genus (Fig 2C and S2 Spreadsheet). After executing these four different computational approaches, consequently the four different constructed MetGEMs networks are achieved (Table 1). In order to further process MetGEMs networks, we then implemented them as reference databases in connection to other data i.e. Amplicon Sequence Variants (ASVs) and taxonomic grouping of ASVs and 16S rRNA gene sequences into the MetGEMs toolbox. Fig 1 demonstrates how MetGEMs toolbox is implemented, whereas the full details can be seen in section of implementation of the MetGEMs toolbox.

## The validated MetGEMs toolbox's prediction capability with shotgun metagenomic sequencing data

After the implementation of the MetGEMs toolbox, it was then validated for its prediction capability using shotgun metagenomic sequencing data from the HMP's gut microbiome (S3 Spreadsheet). We initially assessed the 16S rRNA gene sequence data in all subjects which resulted in a total of 1,762 ASVs, which covered 43 genera and 51 species of bacteria. These ASVs were then mapped to the MetGEMs networks using the MetGEMs toolbox. The results showed that 29 out of 43 genera and 42 out of 51 species were mapped with the MetGEMs networks. The MetGEMs toolbox was then further used to predict the different KO IDs and EC numbers along with individual MetGEMs network. As shown in Table 2, the Pan-Function predicted the largest numbers of KO IDs (1,217.0 ± 41.9) and EC numbers (837.5 ± 33.4). The Core-Function and Core-Weight-Function predicted slightly smaller number of KO IDs and EC numbers for 1,094.5 ± 43.8 and 747.0 ± 31.8, respectively (Table 2). These results indeed

**Table 1. The coverage list of KO IDs and EC numbers identified in MetGEMs networks underlying four different computational approaches.**

| Computational approaches | KO IDs | EC numbers |
|---|---:|---:|
| Pan-Function | 1,470 | 983 |
| Core-Function | 1,459 | 968 |
| Pan-Weight-Function | 1,468 | 980 |
| Core-Weight-Function | 1,459 | 968 |

**Table 2. The coverage list of predicted KO IDs and EC numbers identified by MetGEMs toolbox.**

| Computational approaches | KO IDs | EC numbers |
|---|---:|---:|
| Pan-Function | 1,217.0 ± 41.9 | 837.5 ± 33.4 |
| Core-Function | 1,094.5 ± 43.8 | 747.0 ± 31.8 |
| Pan-Weight-Function | 1,192.5 ± 36.0 | 816.5 ± 26.9 |
| Core-Weight-Function | 1,094.5 ± 43.8 | 747.0 ± 31.8 |

Note: The median ± SD of predicted functions in context of KO IDs and EC numbers.

show that the Core-Function not only could capture distinctive metabolic functions in individual genus (Fig 2C) in the MetGEMs network, but also covered all metabolic functions in the MetGEMs toolbox (Table 2).

To further validate our MetGEMs toolbox, the above predicted results were compared with the corresponding shotgun metagenomic sequencing data using the Spearman's correlation coefficient (SCC). As illustrated in Fig 3A, the results clearly show that the MetGEMs toolbox using Core-Function achieves significantly higher performance among all predictions (p-value <0.05). A high SCC of KO IDs (0.54) and EC numbers (0.68) were observed. It is worth to note that EC numbers prediction has a higher Spearman's correlation coefficient than the KO IDs. This might be possible because the EC numbers are the standard format for GEMs development. In addition, KO IDs are often not 1:1 matches with the EC numbers. To further improve the performance of the MetGEMs toolbox, a taxonomic weight from the HMP's gut microbiome was applied. Surprisingly, as displayed in Fig 3A, the Core-Weight-Function (0.67) and Pan-Weight-Function (0.66) show a slightly lower SCC for the EC numbers prediction. These results suggest that the taxonomic weight did not enhance the performance of MetGEMs toolbox in a meaningful way. Accordingly, this result shows consistency with Kaehler et al. [27] in which taxonomic weight did not improve a prediction performance indeed. Nonetheless, their study supports that taxonomic weight might be considered for further metagenomics-based analysis. Hereby, Core-Function was selected in our case study for MetGEMs toolbox application in a following section.

When comparing the MetGEMs toolbox performance with the other functional prediction tools e.g., PICRUSt2, the results clearly show that the MetGEMs toolbox has higher SCC than PICRUSt2 (0.54 for EC numbers) (Fig 3A). This demonstrates that the MetGEMs toolbox showed a very good performance at capturing metabolic functions. To estimate the null distribution in MetGEMs toolbox for metabolic function inferences, we generated 100 permutation datasets by shuffling ASVs labels. For each permuted data, 200 samples were drawn with replacement to be processed by MetGEMs toolbox, which were used to generate the null distribution of the SCC (S2 Fig) from the bootstrap sampling. The result shows that the SCC is higher than the null distribution and supports the notion that MetGEMs toolbox performed better than random permuted data.

## Using MetGEMs toolbox for assigning enzyme functions and relevant functional categories related human gut microbiome

To apply MetGEMs toolbox, we firstly investigated the prediction capability on the presence of enzyme functions in a form of KO IDs at different functional categories by comparing with corresponding functions from shotgun metagenomic sequencing data. As the results, 773 out of 998 KO IDs (77.5%) are in agreement between the MetGEMs toolbox and shotgun metagenomic sequencing data. This suggests that the MetGEMs toolbox could be further used for assigning enzyme functions in the gut microbiome. As expected, the numbers of predicted KO

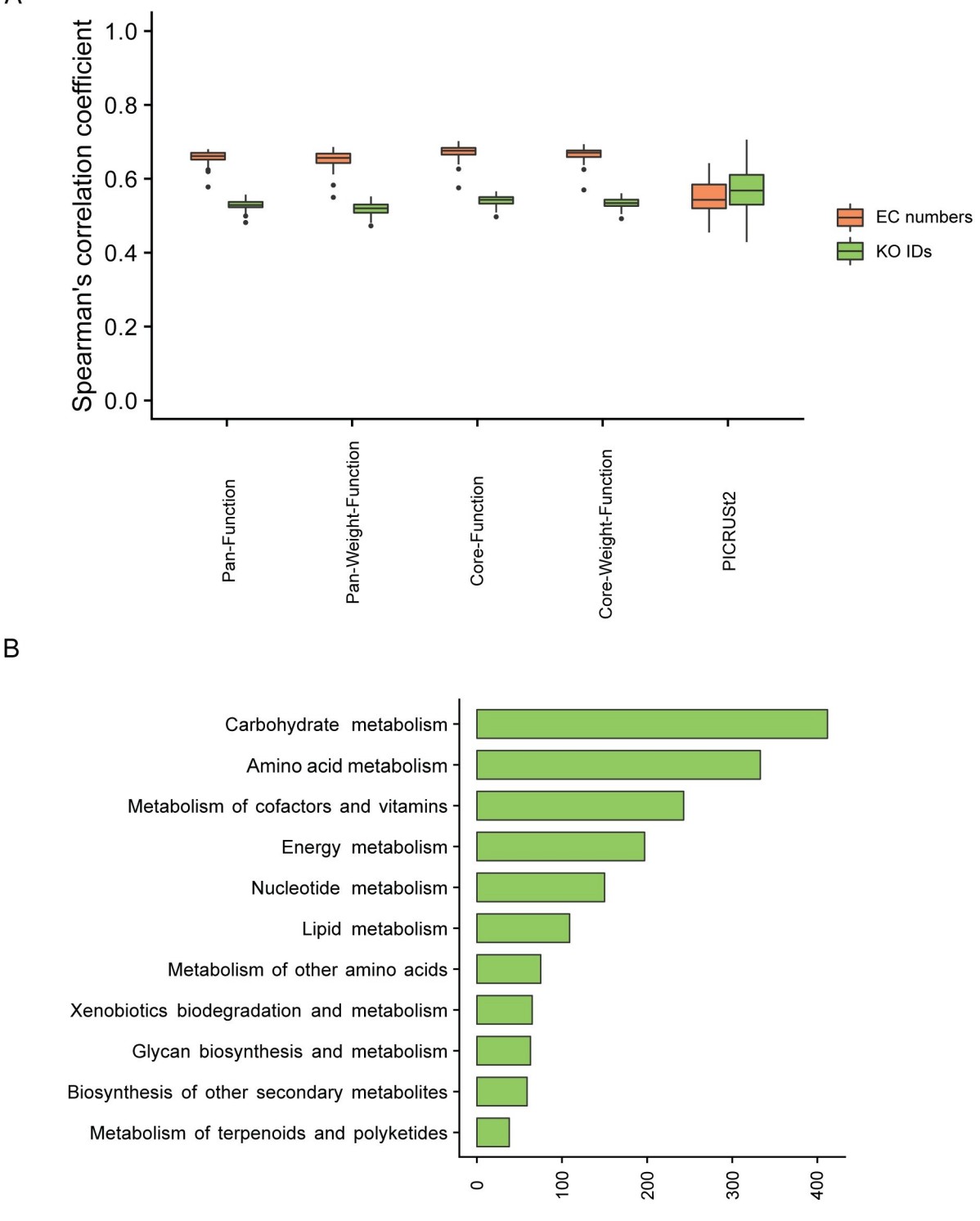

**Fig 3. Validated results of MetGEMs toolbox and its prediction capability for metabolic function inferences.** (A) A Spearman's correlation coefficient (SCC) graph between MetGEMs toolbox's predicted results compared with shotgun metagenomic results of corresponding samples. Note: A correlation coefficient was analyzed for both predicted KO IDs and EC numbers with 61 selected samples. (B) A horizontal bar chart of predicted KO IDs and functionally metabolic categorized by MetGEMs toolbox.

IDs were mostly assigned to three key functional categories related to the human gut micro-biome (Fig 3B), such as carbohydrate metabolism, amino acid metabolism, and metabolism of cofactors and vitamins [28–30].

### Annotating putative enzyme functions related in allergic disease using the MetGEMs toolbox

Investigating the association between the abundance of enzymes from bacterial communities and related diseases is one of the most important goals of microbiome studies [31, 32]. Here, the MetGEMs toolbox with Core-Function was used to detect differentially abundant enzyme functions between healthy and disease groups. In our case study, we firstly profiled 60 infants (39 healthy samples, "controls") and 21 atopic dermatitis samples from subjects of 9 and 19 months infants (S4 Spreadsheet). Afterwards, the 16S rRNA gene sequences were analyzed for identifying significant microbial features (See Materials and Methods). Taxonomic profiles of both healthy and atopic dermatitis samples showed a high number of *Lachnospiraceae* and *Bifidobacteriaceae* (S3 Fig) which is a consistent pattern for Asian children [33]. Subsequently, we used the MetGEMs toolbox to the ASVs abundances and the taxonomic profiles to predict the enzyme functions in the context of KO IDs and EC numbers. The enzyme functions were finally examined by their overall abundance (S4 Spreadsheet). As shown in Fig 4A and 4B, the top five hits of KO IDs and EC numbers are significantly different between healthy and atopic dermatitis samples. We found triosephosphate isomerase (K01803), undecaprenyl-diphospha-tase (K06153), ribulose-phosphate 3-epimerase (K01783) to be significantly higher abundance in atopic dermatitis than healthy samples (Wilcoxon rank-sum test p-value <0.01). Consis-tently, undecaprenyl-diphosphate phosphatase (EC: 3.6.1.27), triose-phosphate isomerase (EC: 5.1.3.1), aspartate carbamoyltransferase (EC: 2.1.3.2), amidophosphoribosyltransferase (EC: 2.4.2.14), triose-phosphate isomerase (EC: 5.3.1.1) were also found to be significantly higher in abundance in atopic dermatitis than healthy samples. When using the MetGEMs toolbox for predicting the difference in metabolic routes between atopic dermatitis and healthy samples, we annotated top five hits of metabolic routes as illustrated in Fig 4C. We identified L-arginine biosynthesis III (via N-acetyl-L-citrulline) (PWY-5154) and L-ornithine biosynthesis I (GLU-TORN-PWY) with higher abundance of atopic dermatitis than healthy samples (Fig 4C) (Wil-coxon rank-sum test p-value <0.01). These results suggest that the MetGEMs toolbox can provide the two putative metabolic routes-associated with atopic dermatitis. As illustrated in Fig 5, the metabolic route visualization of L-arginine biosynthesis III (via N-acetyl-L-citrulline) show that most of enzymes in this route have a higher relative abundance in atopic dermatitis than the healthy samples. These results are consistent with the previous studies where the amino acids alterations were associated with host's phenotype by regulating host's immune system [34, 35].

### Conclusions

The MetGEMs toolbox was developed to use the GEMs as references to infer metagenomic content from 16S rRNA gene sequences with overall focus on annotating metabolic function of human gut microbiome. During our validation, the MetGEMs toolbox's prediction capabil-ity of KO IDs and EC numbers was in good agreement with shotgun metagenomic data. More-over, the MetGEMs toolbox showed better performance on metabolic function inferences than commonly used tools, e.g., PICRUSt2. The MetGEMs toolbox therefore was applied for annotating putative enzyme functions and metabolic routes related to atopic dermatitis in infants. MetGEMs toolbox can further apply with Shotgun datasets and serves as a versatile toolbox for different types of microbiome data.

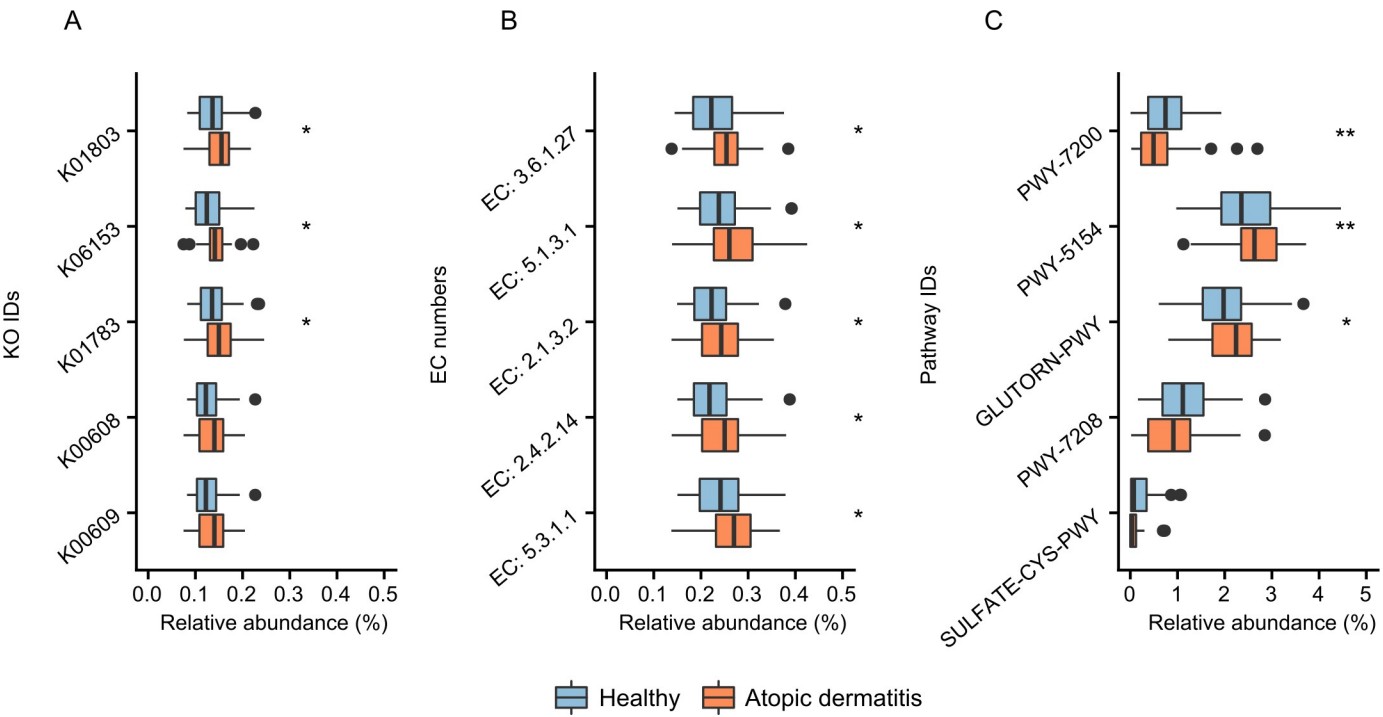

**Fig 4. MetGEMs toolbox predicted metabolic functions and routes from Thai population-based allergy birth cohort study.** Note: Boxplot represents relative abundances of enzyme functions predictions (KO IDs and EC numbers) between healthy and atopic dermatitis samples during 9–19 months. Wilcoxon rank-sum test is used for statistical significance. * and ** correspond to p-value < 0.01 and p-value< 0.005, respectively. (A) The predicted KO IDs including triosephosphate isomerase (K01803), undecaprenyl-diphosphatase (K06153), ribulose-phosphate 3-epimerase (K01783), aspartate carbamoyltransferase (K00608), aspartate carbamoyltransferase catalytic subunit (K00609). (B) The predicted EC numbers including undecaprenyl-diphosphate phosphatase (EC: 3.6.1.27), triose-phosphate isomerase (EC: 5.1.3.1), aspartate carbamoyltransferase (EC: 2.1.3.2), amidophoshoribosyltransferase (EC: 2.4.2.14), triose-phosphate isomerase (EC: 5.3.1.1) (C) The predicted metabolic routes including Superpathway of pyrimidine deoxyribonucleoside salvage (PWY-7200), L-arginine biosynthesis III (via N-acetyl-L-citrulline) (PWY-5154), L-ornithine biosynthesis I (GLUTORN-PWY), Superpathway of pyrimidine nucleobases salvage (PWY-7208), and Superpathway of sulfate assimilation and cysteine biosynthesis (SULFATE-CYS-PWY).

## Materials and methods

An overview of the steps implemented here for MetGEMs toolbox development is depicted in S4 Fig and described in S1 Doc. Details are described in the following sections.

### Ethics statement

This study used the stool samples from the population-based birth cohort study, conducted at Chulalongkorn University, Bangkok, Thailand. The study was approved by the Ethics Committee of King Chulalongkorn Memorial Hospital, Bangkok, Thailand, under the approval reference number 358/58. Parents were provided with sufficient information about our study and agreed to participate in the study. Written informed consent was obtained from the parents or guardians of the participants before collecting clinical data and stool samples. This research was performed according to the Helsinki Guidelines.

### Reference database construction for MetGEMs toolbox

Initially, we collected 818 GEMs and their corresponding metadata (i.e. Reaction IDs associated with KO IDs and EC numbers) from AGORA collections (www.vmh.life). Afterwards, each of GEM out of 818 GEMs was searched for its related genome sequence from PATRIC database [36], and NCBI database (www.ncbi.nlm.nih.gov). Once genome sequences related

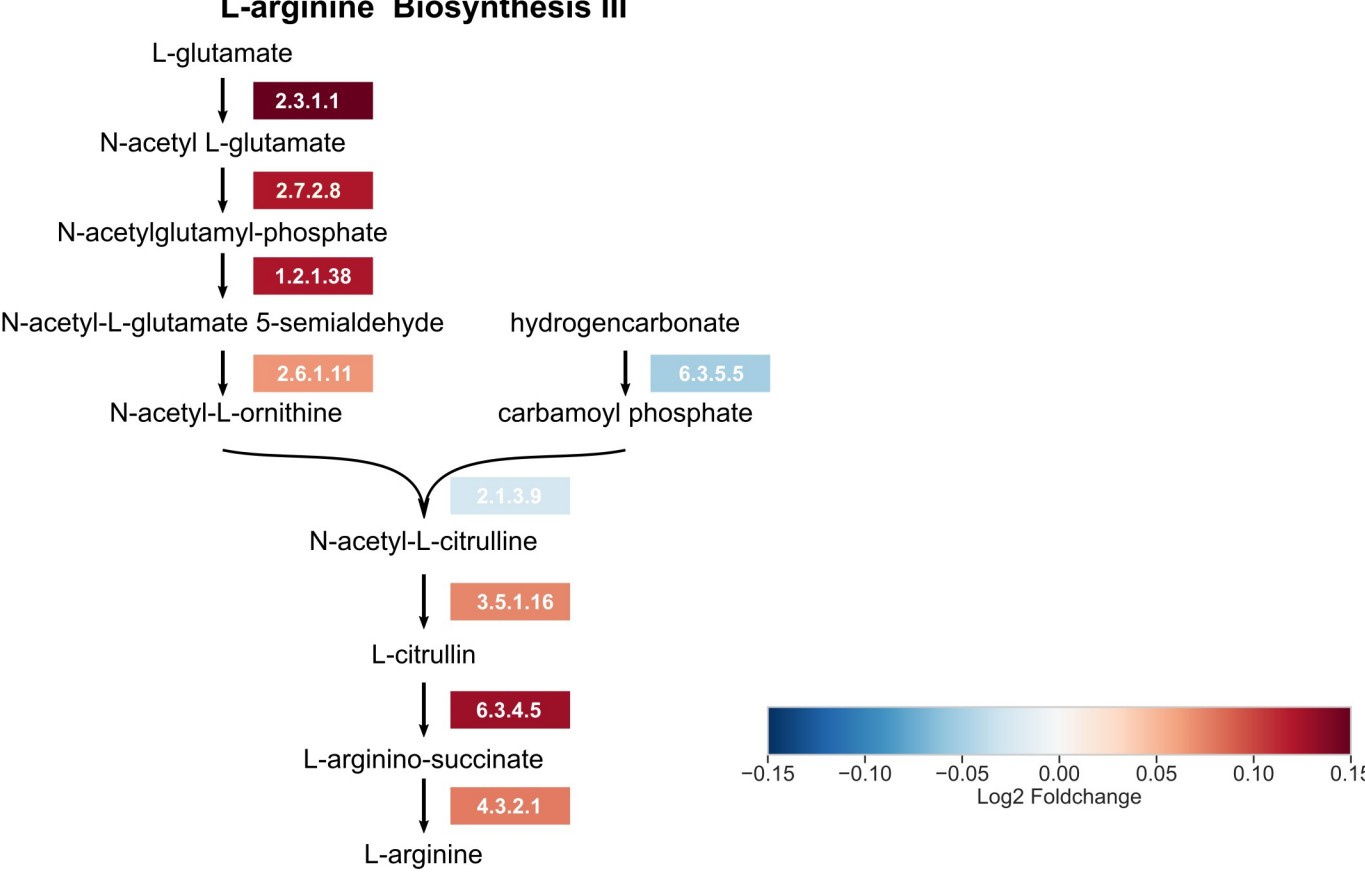

**Fig 5. MetGEMs toolbox identified putative enzyme functions involved in L-arginine biosynthesis III (via N-acetyl-L-citrulline) associated with atopic dermatitis.** Note: Log2 Foldchange shows the ratio of abundance difference of putative enzymes (EC numbers) between atopic dermatitis and healthy samples. Red color represents that atopic dermatitis samples have higher relative abundance of EC numbers than healthy samples. Blue color represents that healthy samples have higher relative abundance of EC numbers than atopic dermatitis samples.

GEMs totally gathered, a reference database called MetGEMs network was then constructed. To explore each MetGEMs network, it was designed through inferencing metabolic functional abundance of the certain taxonomic organism and/or group. Thus, a preliminary construction of individual MetGEMs network using KO IDs, EC numbers, and biochemical reactions across certain genus and/or specific species, which achieved from AGORA collections, was performed. In order to obtain MetGEMs network, four different computational approaches, i.e. Pan-Function, Core-Function, Pan-Weight-Function, and Core-Weight-Function, were used.

Under Pan-Function, a computation was based on average of numbers of KO IDs and EC numbers across all organisms used in the study. A computation by Pan-Function is formulated as Eq (1) and Eq (2).

$$fq(TG_i, KO_j) = \sum_{G_k \in TG_i} fq(G_k, KO_j) \tag{1}$$

$$fq(TG_i, EC_j) = \sum_{G_k \in TG_i} fq(G_k, EC_j) \tag{2}$$

In contrast, a computation of Core-Function means intersection of numbers of KO IDs and EC numbers across all organisms used in the study. A computation by Core-Function is

formulated as Eq (3) and Eq (4).

$$fq(TG_i, KO_j) = \begin{cases} 1 \ \forall fq(G_k, KO_j) \geq 1 \ for \ G_k \in TG_i \\ 0 \ otherwise \end{cases} \tag{3}$$

$$fq(TG_i, EC_j) = \begin{cases} 1 \ \forall fq(G_k, EC_j) \geq 1 \ for \ G_k \in TG_i \\ 0 \ otherwise \end{cases} \tag{4}$$

TG is Taxonomic Group, $fq(TG_i, KO_j)$ is an abundance from Taxonomic Group i and KO IDs j. $fq(TG_i, EC_j)$ is an abundance from Taxonomic Group i and EC number j. G represents a GEM. $fq(G_k, KO_j)$ is an abundance of $KO_j$ from $G_k$. $fq(G_k, EC_j)$ is an abundance of $EC_j$ from $G_k$.

For the other two computational functions, they were relied on taxonomic weighting of each organism. Pan-Weight-Function was used to compute average of numbers of KO IDs and EC numbers across taxonomic weighting of all organisms used. In this study, the average of species abundance in HMP gut's microbiome was used as a taxonomic weighting. While Core-Weight-Function was also similar with Pan-Weight-Function in context of taxonomic weighting, however, a computation was considered by intersection of numbers of KO IDs and EC numbers instead across all organisms. A computation by Pan-Weight- and Core-Weight-Function are formulated as Eq (5) and Eq (6).

$$fq(wG_k, KO_j) = fq(w_k) \times fq(G_k, KO_j) \tag{5}$$

$$\boldsymbol{fq(wG_k, EC_j) = fq(w_k) \times fq(G_k, EC_j)} \tag{6}$$

Where the $\boldsymbol{fq(wG_k, KO_j)}$ and $\boldsymbol{fq(wG_k, EC_j)}$ are then replacing $\boldsymbol{fq(G_k, KO_j)}, \boldsymbol{fq(G_k, EC_j)}$ in the unweight versions (Eqs 1–4). If all weights $\boldsymbol{fq(w_k)}$ is equal to 1, then the weight version is equivalent to the unweight version.

## MetGEMs toolbox's implementation

MetGEMs toolbox took three input data 1) MetGEMs network as a reference database, 2) a tab-delimited ASV-sample table, and 3) a mapping data containing taxonomic group for each ASVs of 16S rRNA gene sequence data. An implementation of MetGEMs toolbox is described as follows.

**Step 1: Converting ASV-sample table into Taxonomic Group-sample table.**   An ASV-sample table was converted into Taxonomic Group-sample table by summing the frequencies of ASV with the same taxonomy group, as follows as Eq (7).

$$fq(TG_i, Sample_j) = \sum_{ASV_k \in TG_i} fq(ASV_k, Sample_j) \tag{7}$$

Here, TG is Taxonomic Group, and $fq(TG_i, Sample_j)$ is an abundance from Taxonomic Group i and Sample j in Taxonomic Group-sample table.

**Step 2: Normalizing taxonomic grouping abundance.**   Due to the ASV abundance reflects 16S rRNA full number of occurrences (e.g., sequence count or relative abundance) of organisms assigned to a given ASV. To implement organismal abundance, we thus performed normalizing taxonomic grouping abundance. Here, we divided the ASVs' abundance by the putative 16S rRNA gene copy number, which was defined as the median of 16S rRNA gene copy number of the organisms pooled at the ASV's taxonomic grouping.

**Step 3: Converting a taxonomic grouping-sample table into a metabolic function-sample table.** We derived a KO-sample and EC-sample tables by combining MetGEMs network corresponding to the taxonomic grouping abundance in the sample: The formula shows as Eq (8) and Eq (9).

$$fq(KO_i, Sample_j) = \sum_{TaxonomicGroup_k} fq(TG_k, Sample_j) \times fq(TG_k, KO_i) \tag{8}$$

$$fq(EC_i, Sample_j) = \sum_{TaxonomicGroup_k} fq(TG_k, Sample_j) \times fq(TG_k, EC_i) \tag{9}$$

Hereby, (TG, KO) and (TG, EC) are the MetGEMs networks of KO IDs and EC numbers, respectively.

## Testing and validation of MetGEMs toolbox with shotgun metagenomic sequencing data

In order to test and validate MetGEMs toolbox, we divided into five steps as follows.

**Step 1: Preparing input data.** All human feces data that have 16S rRNA gene sequence data (e.g., ASV sequence) and corresponding shotgun metagenome sequencing data were downloaded from NCBI database according to the HMP website (http://hmpdacc.org/).

**Step 2: Predicting metabolic functions based on 16S rRNA gene sequence data.** To prepare 16S rRNA sequence data for MetGEMs toolbox, the 16S rRNA gene sequences were processed by BBDUK (v.38.12) with an option *ktrim = l qtrim = r trimq = 15 minlength = 150* to trim adapter sequences (using adapter sequences as reported by Huttenhower et al. [25]) and filter low quality reads. The quality sequences were further processed with DADA2 (v1.6.0) [37] into ASVs with default options. The taxonomy assignment of ASVs was then done with QIIME2 (v2019.1) using *feature-classifier classify-sklearn* [38] with Greengenes database v13.8 [39]. These processes produced ASV-sample table and mapping file containing taxonomic grouping for each ASVs of 16S rRNA gene sequence data and further used for metabolic function prediction within MetGEMs toolbox.

**Step 3: Generating reference metabolic function profile from shotgun metagenomic sequencing data.** To prepare shotgun metagenomic sequencing data as a reference metabolic function profile for validation, HUMAnN2 (v0.11.2) [7] was used. Initially, each sample of forward and reverse reads were concatenated into the same file. HUMAnN2 was then run to identify the abundances of annotated UniRef90 gene families using–*search-mode uniref90 and default cut-off parameters*. The abundances of UniRef90 gene families were regrouped into KO IDs and EC numbers using *humann2_regroup_table*.

**Step 4: Validating MetGEMs toolbox with shotgun metagenomic sequencing data.** To validate how MetGEMs toolbox results predicted with shotgun metagenomic sequencing data, a statistical measure of the strength of a correlation analysis between paired data was performed. Here, SCC was chosen and computed to compare between MetGEMs predicted results and reference metabolic function profile from shotgun metagenomic sequencing data for each sample.

**Step 5: Statistical test of MetGEMs toolbox.** For estimating the significance of MetGEMs toolbox prediction, the permutation analysis together with bootstrap sampling was used. To perform, 100 permutation datasets were generated by shuffling ASVs labels in the ASV-sample table. For each permuted data, 200 samples were drawn with replacement to be processed by MetGEMs toolbox. To the end, the null distribution was then computed from the bootstrap sampling and compared with the results from Step 4 (S2 Fig).

## Metabolic function inference of microbiome associated in allergic disease by MetGEMs toolbox

To use the developed MetGEMs toolbox, we next assessed MetGEMs capabilities for predicting metagenomes based on our bacterial amplicon sequencing of the 16S rRNA gene sequence data from Thailand Pediatric Allergy Research Cohort. Hereby, we divided into three steps as follows:

**Step 1: Preparing input data.**   Fecal samples were collected from 60 infants who participated in a population-based allergy birth cohort study at King Chulalongkorn Memorial Hospital, Bangkok, Thailand. These included 39 healthy and 21 atopic dermatitis samples. Notably, atopic dermatitis was diagnosed by a pediatric allergist according to the criteria of the American Academy of Dermatology [40]. The study was approved by the Ethics Committee of King Chulalongkorn Memorial Hospital, Bangkok, Thailand, under the approval reference number 358/58. Written informed consent was obtained from the parents or guardians of the participants. Information on age, gender, family history, pet, and mode of delivery are collected as shown in S4 Spreadsheet. The stool samples were collected from 9–19 months of ages and frozen at -80˚C until DNA extraction. The DNA samples were then subjected to amplification with 341f and 805r primers with Illumina sequencing adapter and then sequencing with Illumina MiSeq System (Illumina, San Diego, CA). Out of 60 infants, the sequencing dataset comprised of 48 samples (31 healthy and 17 atopic dermatitis samples) at 9 months, and 53 samples (36 healthy and 17 atopic dermatitis samples) at 19 months (S4 Spreadsheet).

**Step 2: Data processing of 16S rRNA gene sequence data towards predicting metabolic functions.**   We processed the sequencing read dataset using the same parameters and tools as in previous section. In brief, the sequencing reads datasets were trimmed and filtered with BBDuk (v.38.12) and then denoised with DADA2 (v1.6.0) into ASVs. These ASVs were then assigned with taxonomy using QIIME2's taxonomy classifier with Greengenes (v13.8) [39] trained on 341f-805r region. After that, MetGEMs toolbox with Core-Function was used to investigate the overall metabolic functions across all samples, hereby KO IDs and EC numbers in each sample were rank-transformed and the geometric means of KO IDs and EC numbers of each condition (i.e. atopic dermatitis and healthy samples) were computed. To detect a difference of KO IDs and EC numbers abundance between atopic dermatitis and healthy samples, the Wilcoxon rank-sum test was used.

**Step 3: Mapping predicted functions on metabolic routes.**   To map predicted functions on metabolic routes, MinPath (v1.2) [41] was used to select the minimum set of routes according to the availability of EC numbers in each sample. EC numbers were firstly grouped according to the selected routes, and the route abundance was then computed as the harmonic mean of EC numbers abundance within the selected routes. Wilcoxon rank-sum test was used to find a difference of relative abundance route between atopic dermatitis and healthy samples.

## Supporting information

**S1 Fig. Scatter plot of KO IDs and EC numbers from 818 GEMs in AGORA collections.** (PDF)

**S2 Fig. Spearman's correlation coefficient of MetGEMs results with the corresponding shotgun metagenomic sequencing.** Comparison are shown between results from real dataset and null distribution from permutation and bootstrap sampling. Independent T-test was used to assess the differences of (A) KO IDs prediction (B) EC numbers prediction. (PDF)

**S3 Fig. Relative abundance of bacterial families of human gut microbiome from Thailand Pediatric Allergy Research Cohort.**
(PDF)

**S4 Fig. Overall diagram of MetGEMs toolbox development.**
(PDF)

**S1 Spreadsheet. Number of genome-scale models used to build MetGEMs network (genus level) and MetGEMs network (species level).**
(XLSX)

**S2 Spreadsheet. Function distribution of AGORA collections at class level.**
(XLSX)

**S3 Spreadsheet. List of HMP Sample IDs.**
(XLSX)

**S4 Spreadsheet. MetGEMs toolbox prediction on Thailand Pediatric Allergy Research Cohort.**
(XLSX)

**S1 Doc. Supplementary Methods of MetGEMs toolbox development.**
(PDF)

## Acknowledgments

We would like to thank Professor Yong Poovorawan, Dr. Nasamon Wanlapakorn, from Center of Excellence Clinical Virology, Faculty of Medicine, Chulalongkorn University, Bangkok, Thailand, Paediatric allergists and research team from Division of Allergy and Immunology, Department of Paediatrics, Faculty of Medicine, King Chulalongkorn Memorial Hospital, Bangkok, Thailand for enrollment and follow-up the subjects in this cohort. We also would like to acknowledge Kasetsart University Research and Development Institute (KURDI) at Kasetsart University, Leibniz Institute for Natural Product Research and Infection Biology, Germany, Computational Biomodelling Laboratory for Agricultural Science and Technology (CBLAST), Faculty of Science and Omics Center for Agriculture, Bioresources, Food, and Health, Kasetsart University (OmiKU). In addition, we also would like to thank Chakkapan Sapkaew, Amorntep Kingkaw, and Nachon Raethong for their assistance on fruitful discussion. PP would like to thank Interdisciplinary Graduate Program in Bioscience, Faculty of Science, and the International Affairs Division (IAD) at Kasetsart University. WV would like to acknowledge Department of Zoology and International SciKU Branding (ISB), Faculty of Science, Kasetsart University.

## Author Contributions

**Conceptualization:** Gianni Panagiotou, Wanwipa Vongsangnak.

**Data curation:** Preecha Patumcharoenpol, Massalin Nakphaichit, Anchalee Senavonge.

**Formal analysis:** Preecha Patumcharoenpol, Massalin Nakphaichit.

**Funding acquisition:** Preecha Patumcharoenpol, Gianni Panagiotou, Narissara Suratannon, Wanwipa Vongsangnak.

**Investigation:** Narissara Suratannon, Wanwipa Vongsangnak.

**Methodology:** Massalin Nakphaichit, Anchalee Senavonge, Narissara Suratannon, Wanwipa Vongsangnak.

**Project administration:** Narissara Suratannon, Wanwipa Vongsangnak.

**Resources:** Gianni Panagiotou, Narissara Suratannon, Wanwipa Vongsangnak.

**Software:** Gianni Panagiotou, Wanwipa Vongsangnak.

**Supervision:** Wanwipa Vongsangnak.

**Validation:** Preecha Patumcharoenpol, Wanwipa Vongsangnak.

**Visualization:** Preecha Patumcharoenpol.

**Writing – original draft:** Preecha Patumcharoenpol, Wanwipa Vongsangnak.

**Writing – review & editing:** Preecha Patumcharoenpol, Massalin Nakphaichit, Gianni Panagiotou, Anchalee Senavonge, Narissara Suratannon, Wanwipa Vongsangnak.

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
