## [Decision Letter · Decision Letter 0]

31 Aug 2020

Dear Prof. Vongsangnak,

Thank you very much for submitting your manuscript "MetGEMs Toolbox: Metagenome-scale models as integrative toolbox for uncovering metabolic functions and routes of human gut microbiome" for consideration at PLOS Computational Biology.

As with all papers reviewed by the journal, your manuscript was reviewed by members of the editorial board and by several independent reviewers. In light of the reviews (below this email), we would like to invite the resubmission of a significantly-revised version that takes into account the reviewers' comments.

Ensure that your revisions offer a point-by-point response. We also expect the revision to provide considerably more methodological clarity and detail, and a proper placement of the proposed methodology in the context of existing work.

We cannot make any decision about publication until we have seen the revised manuscript and your response to the reviewers' comments. Your revised manuscript is also likely to be sent to reviewers for further evaluation.

Sincerely,

Sergei L. Kosakovsky Pond, PhD

Associate Editor

PLOS Computational Biology

Mark Alber

Deputy Editor

PLOS Computational Biology

Reviewer's Responses to Questions

**Comments to the Authors:**

Reviewer #1: This work proposes the MetGEMs as integrative toolbox which can predict the functional capabilities of microbiome from amplicon sequencing data. The proposed method improved over the existing method by applying genome-scale metabolic model as a reference instead of draft genomes which results in the functional prediction improvement. However, there are the comments that are needed to be addressed in the manuscript before publication.

1. There are several functional predictors publicly available, the authors should state the reasons for selection of PICRUSt2?

2. Can the authors discuss for further improvement of MetGEMs e.g., can it be applied by Shotgun datasets?

3. The toolbox seems to be limited with the gut microbiome data or not.

4. Can the authors compare the metabolic results between Pan-, Pan-Weight, Core-, Core-Weight? Which one is the suitable for further functional prediction application?

5. For the better perfume the MetGEMs, is possible to include data in the GitHub as well?

6. Language is needed to be improved, such as “However, there have not yet been used genome-scale models (GEMs) as scaffold toolbox for microbiome analysis at a systematic level.” in the introduction part.

Reviewer #2: In their paper “MetGEMs Toolbox: Metagenome-scale models as integrative toolbox for uncovering metabolic functions and routes of human gut microbiome”, Patumcharoenpol et al. present MetGEMs, a tool for inferring metagenome-wide KEGG Ortholog and Enzyme Commission number abundances by mapping amplicon sequencing data to genome-scale metabolic network reconstructions. They demonstrate an improvement in performance relative to PICRUSt2 based on increased correlation between inferred KO/EC abundances from amplicon sequencing data and paired metagenomics data. While the software appears to be well-written, the methods underlying the software and all of the benchmarking analyses contained in the paper are severely under-described. Increasing the level of detail in the manuscript is very critical to engaging new users of the software, especially because there appears to be no formal documentation for the software in the GitHub repository. Overall, I feel that this is a very valuable contribution, but it will have much higher impact and probability of adoption if the quality and detail of the manuscript is improved.

Major comments

The authors statements of novelty for their method are overstated and should be made more nuanced. Specifically, the authors seem to be communicating that inference of community-level metabolic functionality from 16S rRNA gene sequencing data has never been performed. For example, in lines 70-72, the authors state the AGORA GEMs “... have not yet been used as a scaffold toolbox to infer metagenomic content from 16S rRNA sequenced samples at a systematic level”, but this was done to a small extent in the original AGORA paper (see the last figure in Magnusdottir et al. 2017 for inference of reaction content from both 16 rRNA gene sequencing data and shotgun metagenomics data). The authors of the AGORA paper have also since performed more in-depth versions of this analysis on many occasions (for example, see Baldini et al., BMC Biology 2020, “Parkinson’s disease-associated alterations of the gut microbiome predict disease-relevant changes in metabolic functions”). The authors’ statements describing the need for their method or the innovative aspects should include this context. In my opinion, the true innovation here is that the methodology for performing this analysis is open and packaged as software, whereas previous approaches in the field have not shared code and use obscure methods that would be difficult to reproduce. In other words, the authors have developed a tool for actually performing this analysis.

Methods are described with insufficient detail for the majority of analyses in the paper. We recommend adding more detail to the description of every analysis (including steps of the MetGEM pipeline as well as all of the analyses used to test MetGEM performance). Here is a non-exhaustive list of examples of issues that I cannot find answers to in the manuscript: 1) In the atopic dermatitis analysis, was Core- or Pan-functionality used to compute KO/ECs? 2) No methods are provided for 16S rRNA gene sequence data processing other than the software and databases used. BBDUK, DADA2, and QIIME2 all have many parameters selected by the user, and the authors provide zero parameter values. This needs much more detail to be reproducible. 3) Similar to issue (2), no details for parameters used in HUMAnN2 are provided. 4) Insufficient detail on bootstrapping is provided. Both the number of samples and the *size* of each sample are needed. Furthermore, in the analysis in Fig S1, I can’t tell what the non-bootstrapped line is supposed to be indicating; the authors should describe this.

Minor comments

Figure 2A--what do the green and blue lines represent? It looks like number of reactions, but it isn’t labelled anywhere. Also, the choice to annotate the figure with the correlation coefficient is very odd--it would be clearer to simply add an additional scatter plot showing the data that went into the correlation, and the goodness of fit for the correlation.

Within the results around lines 127-159, no justification is given for the “weighted” versions of the core/pan metrics, and it’s not clear how they are computed. The use of these metrics should be justified when they are introduced here, and their computation should be briefly described (with more detail in the methods).

Figure 1, “Specie” is not the singular form for “species”--both singular and plural forms of the words are “species”. This is unfortunate and confusing, but the authors should correct “Specie” to “Species” to be consistent with English conventions.

References to the 16S rRNA gene sequence should use the complete terminology, e.g., “16S rRNA gene sequencing”, and not “16S rRNA sequencing”, since the *gene*

**Have all data underlying the figures and results presented in the manuscript been provided?**

Reviewer #1: None

Reviewer #2: Yes

PLOS authors have the option to publish the peer review history of their article (what does this mean?). If published, this will include your full peer review and any attached files.

Reviewer #1: No

Reviewer #2: No
---

## [Decision Letter · Decision Letter 1]

3 Nov 2020

Dear Prof. Vongsangnak,

We are pleased to inform you that your manuscript 'MetGEMs Toolbox: Metagenome-scale models as integrative toolbox for uncovering metabolic functions and routes of human gut microbiome' has been provisionally accepted for publication in PLOS Computational Biology.

Best regards,

Sergei L. Kosakovsky Pond, PhD

Associate Editor

PLOS Computational Biology

Mark Alber

Deputy Editor

PLOS Computational Biology

Reviewer's Responses to Questions

**Comments to the Authors:**

Reviewer #1: The authors have solved my concerns, and thus I think that the current verison can be accepted by PLOSCB.

Reviewer #2: The authors have comprehensively addressed my concerns and I look forward to this tool being used in the field.

**Have all data underlying the figures and results presented in the manuscript been provided?**

Reviewer #1: None

Reviewer #2: Yes

PLOS authors have the option to publish the peer review history of their article (what does this mean?). If published, this will include your full peer review and any attached files.

Reviewer #1: No

Reviewer #2: No

---

## [Editor Report · Acceptance letter]

9 Dec 2020

PCOMPBIOL-D-20-00865R1 

MetGEMs Toolbox: Metagenome-scale models as integrative toolbox for uncovering metabolic functions and routes of human gut microbiome

Dear Dr Vongsangnak,

I am pleased to inform you that your manuscript has been formally accepted for publication in PLOS Computational Biology. Your manuscript is now with our production department and you will be notified of the publication date in due course.

With kind regards,

Nicola Davies
